Users’ needs for a digital smoking cessation application and how to address them: A mixed-methods study

Albers Nele 1 n.albers@tudelft.nl
http://orcid.org/0000-0002-8161-5722 Neerincx Mark A. 1 2
http://orcid.org/0000-0002-9758-9004 Penfornis Kristell M. 3
http://orcid.org/0000-0001-8485-7092 Brinkman Willem-Paul 1
1 Department of Intelligent Systems, Delft University of Technology , Delft , The Netherlands
2 Department of Perceptual and Cognitive Systems, Nederlandse Organisatie voor Toegepast Natuurwetenschappelijk Onderzoek (TNO) , Soesterberg , The Netherlands
3 Health, Medical and Neuropsychology Unit, Institute of Psychology, Leiden University , Leiden , The Netherlands
Khosravi Mohsen
Electronic publication date: 2022 Aug 19
Publication date: 2022
Volume: 10
Electronic Location ID: e13824
Received 2022 Apr 1; Accepted 2022 Jul 10
Copyright: © 2022 Albers et al.
Copyright year: 2022
Copyright holder: Albers et al.
License: This is an open access article distributed under the terms of the Creative Commons Attribution License, which permits unrestricted use, distribution, reproduction and adaptation in any medium and for any purpose provided that it is properly attributed. For attribution, the original author(s), title, publication source (PeerJ) and either DOI or URL of the article must be cited.
License URL: https://creativecommons.org/licenses/by/4.0/

Keywords: Smoking cessation, Physical activity, Behavior change, Virtual coach, Conversational agent, Chatbot, eHealth, User needs, Thematic analysis

Funding: Netherlands Organization for Scientific Research (NWO) Commit2Data – Big Data & Health 628.011.211 This work is part of the multidisciplinary research project Perfect Fit, which is supported by several funders organized by the Netherlands Organization for Scientific Research (NWO), program Commit2Data – Big Data & Health (Project Number 628.011.211). The funders had no role in study design, data collection and analysis, decision to publish, or preparation of the manuscript.

==============================
Background

Despite their increasing prevalence and potential, eHealth applications for behavior change suffer from a lack of adherence and from dropout. Advances in virtual coach technology provide new opportunities to improve this. However, these applications still do not always offer what people need. We, therefore, need a better understanding of people’s needs and how to address these, based on both actual experiences of users and their reflections on envisioned scenarios.

Methods

We conducted a longitudinal study in which 671 smokers interacted with a virtual coach in five sessions. The virtual coach assigned them a new preparatory activity for quitting smoking or increasing physical activity in each session. Participants provided feedback on the activity in the next session. After the five sessions, participants were asked to describe barriers and motivators for doing their activities. In addition, they provided their views on videos of scenarios such as receiving motivational messages. To understand users’ needs, we took a mixed-methods approach. This approach triangulated findings from qualitative data, quantitative data, and the literature.

Results

We identified 14 main themes that describe people’s views of their current and future behaviors concerning an eHealth application. These themes relate to the behaviors themselves, the users, other parties involved in a behavior, and the environment. The most prevalent theme was the perceived usefulness of behaviors, especially whether they were informative, helpful, motivating, or encouraging. The timing and intensity of behaviors also mattered. With regards to the users, their perceived importance of and motivation to change, autonomy, and personal characteristics were major themes. Another important role was played by other parties that may be involved in a behavior, such as general practitioners or virtual coaches. Here, the themes of companionableness, accountability, and nature of the other party (i.e., human vs AI) were relevant. The last set of main themes was related to the environment in which a behavior is performed. Prevalent themes were the availability of sufficient time, the presence of prompts and triggers, support from one’s social environment, and the diversity of other environmental factors. We provide recommendations for addressing each theme.

Conclusions

The integrated method of experience-based and envisioning-based needs acquisition with a triangulate analysis provided a comprehensive needs classification (empirically and theoretically grounded). We expect that our themes and recommendations for addressing them will be helpful for designing applications for health behavior change that meet people’s needs. Designers should especially focus on the perceived usefulness of application components. To aid future work, we publish our dataset with user characteristics and 5,074 free-text responses from 671 people.

Introduction

When creating an eHealth application for behavior change, one is confronted with many choices. The first one relates to behavior change techniques, for which Michie et al. (2013) alone formulated 93 options, including coping planning, self-talk, and social support. Second, one has to decide how to implement these behavior change techniques. For example, should users create coping plans regularly, or only when they feel the need? Third, it gets more complicated when another party, such as a virtual coach, general practitioner, or somebody from the social environment, is involved. When should these parties be included, and how? And lastly, all of these choices should be made so that users use and continue to use the application. For people to use an application, it has to meet their needs. So what are users’ needs for using a behavior change application, and what does this imply for somebody creating such an application?

Recent years have seen a surge of eHealth applications with 78,000 new ones in major app stores in 2017 alone (Research2Guidance, 2017). These applications can be easy to use, available at all times, scalable, cost-effective, and can facilitate tailoring of the intervention (Liao et al., 2016). These characteristics make such applications beneficial for people wishing to change their health behavior, which can be difficult without help. For instance, more than two-thirds of adult smokers in the United States want to quit smoking (Babb et al., 2017), but most unassisted quit attempts fail (Cooper et al., 2010). However, despite their potential, users commonly do not adhere to eHealth applications or abandon them entirely (Beun et al., 2016; Greenhalgh et al., 2017; Kelders, Van Zyl & Ludden, 2020). Thus, there appears to be a mismatch between what the applications offer and what users need.

To improve behavior change applications, users’ needs must be better understood. Thereby, it is crucial to take a holistic approach that considers not only the technology itself but also the user and their environment (van Gemert-Pijnen et al., 2011). Previous work in the context of quitting smoking has, for example, found that the intuitiveness of the user interface (Kulhánek et al., 2018), users’ experience with computers (Ghorai & Ray, 2019), the appreciation expressed by a conversational agent (Kulhánek et al., 2018), and support from one’s social environment (Struik et al., 2018) play a role. This illustrates the diversity of factors that may need to be considered in eHealth applications.

Studies for getting input on the system design from users differ in two ways. First, they employ systems of differing maturity, ranging from mere design ideas to complete applications. Each of the two extremes has an advantage: the former allows one to test multiple design options more easily; the latter helps users to more accurately identify benefits and barriers to using the application (Canada Health Infoway, 2013). Second, users interact with a system for varying amounts of time before being asked for their input. Both very short and very long uses are at risk of resulting in an overly positive evaluation of a system: the former because people’s initial curiosity and excitement about a novel system tend to fade as they become more aware of the system’s limitations (Croes & Antheunis, 2021; Sadeghi et al., 2021), and the latter because people for whom the system does not work tend to drop out according to the law of attrition (Eysenbach, 2005). Thus, data should be collected in the middle range, where the novelty effect has worn off and average users have not yet dropped out. This is to allow one to more accurately assess users’ needs.

This study aims to get a more accurate assessment of users’ needs for eHealth applications for behavior change. To this end, we collected data on both the use of an application and views on multiple design ideas from this middle time range. More precisely, we conducted a longitudinal study in which 671 smokers interacted with a text-based virtual coach. Virtual coaches or conversational agents have been receiving a lot of attention in the health context due to their potential ability to increase engagement, provide and discuss relevant information, and form a connection with users (Montenegro, da Costa & da Rosa Righi, 2019). Participants of our study interacted with such a virtual coach in up to five sessions spread over at least 9 days. In each session, participants were assigned a new preparatory activity for quitting smoking or increasing physical activity, with the latter possibly aiding the former (Haasova et al., 2013; Trimbos Instituut, 2016) and vice versa (Papathanasiou et al., 2014). To gain a comprehensive understanding of participants’ needs for using the application, we conducted a mixed-methods analysis. This analysis was based on participants’ characteristics such as their physical activity identity, their feedback on their activities as well as barriers and motivators and thus information on actual behavior, their views on videos of interaction scenarios described after completing the five sessions and thus information on experience-based behavioral intentions for multiple design options, and findings from the literature. We found a comprehensive set of 14 themes that describe users’ needs. We used these themes to formulate recommendations to support designers of future health behavior change applications. To further aid future research on understanding user needs, we publish our data together with this article.

Materials and Methods

We conducted a longitudinal study from 20 May 2021 until 30 June 2021. The Human Research Ethics Committee of Delft University of Technology approved the study (Letter of Approval number: 1523), and we preregistered the study in the Open Science Framework (OSF) (Albers & Brinkman, 2021).

Study design

The study followed a mixed design with five sources of information. We collected the first four from participants: their characteristics (e.g., physical activity identity), their feedback on their previously assigned preparatory activity for quitting smoking or increasing physical activity, barriers and motivators they had for doing their activities, and their views on interaction scenarios for a virtual coach (e.g., whether participants would like to receive motivational messages). The user characteristics were quantitative, the barriers and motivators were qualitative, and the activity feedback and views on interaction scenarios were quantitative and qualitative. Each participant saw a random selection of two interaction scenarios with the goal of presenting each scenario the same number of times across the sample population. Fig. 1 illustrates how we gathered data from participants. Finally, previous studies provided information to triangulate the findings from the other four sources of information. Triangulation of multiple data sources or methods has been described as a way to examine the validity of qualitative research and to obtain a comprehensive understanding of a phenomenon (Nancy Carter, Bryant-Lukosius & Alba DiCenso, 2014). A successful example of triangulating qualitative findings with previous studies as part of the analysis is the work of Nahar et al. (2021) in the context of software engineering, which we took as an inspiration for this study.

Figure 1 Study design.

Design of the study, including the study components, collected data, and participant flow. Icons illustrate the four types of data we collected from participants: characteristics, feedback on preparatory activities, barriers and motivators for doing the activities, and views on interaction scenarios for a virtual coach. The numbers next to the study components indicate how many participants started the respective component. For the post-questionnaire, we show which data we collected from participants who did not complete it.

Materials

We used the online crowdsourcing platform Prolific to recruit, invite and communicate with participants, Qualtrics to host the questionnaires and instructions for the sessions, Google Compute Engine to host the virtual coach and the sessions using Rasa X, and YouTube to host the videos shown for the interaction scenarios.

The virtual coach used for the sessions was implemented in Rasa (Bocklisch et al., 2017) and had the name Sam. Sam introduced itself as wanting to help people to prepare to quit smoking and become more physically active, with the latter possibly facilitating the former. The code of Sam can be accessed online (Albers, 2022). Sam proposed a new preparatory activity related to quitting smoking or increasing physical activity in each session. The virtual coach randomly drew these activities from a pool of 24 activities, 12 each for quitting smoking and increasing physical activity. The activities were based on components of the smoking cessation app StopAdvisor (Michie et al., 2012) and future-self exercises (Meijer et al., 2018; Penfornis, Gebhardt & Meijer, 2021), and reviewed by a psychologist and smoking cessation expert. Examples of activities are formulating a rule for not smoking or tracking one’s physical activity. Table S1 shows the complete list of activities. An example of a conversation with Sam is shown in Fig. S8. Based on their acceptance of Sam measured in the post-questionnaire with six items on scales from −5 to 5 and with 0 being neutral (Albers, Neerincx & Brinkman, 2022), participants had an overall positive attitude toward Sam (M = 2.50, SD = 1.68, 95% HDI = [2.32, 2.68]).

In the post-questionnaire, each participant saw 2 out of 13 interaction scenarios in video form. Each video presented an imaginary persona alongside her situation and described an interaction for this persona. The video ended with a question about whether the viewer would engage in the interaction if they were the persona. The topics for the scenarios (Table S2) were drawn from the literature and discussions within the consortium of the multidisciplinary Perfect Fit project (Meijer et al., 2021) that this study is a part of. This project aims to develop an app that helps smokers quit smoking and become more physically active. There were two versions for each video, one with a male and one with a female persona. Male and female participants saw a video with a persona whose gender matched their own; participants with a different gender saw one with a persona whose gender was chosen randomly. The information in the videos was presented using text and voice-over. Table S3 provides links to the videos on YouTube.

Measures

We used the following measures in our analysis:

Activity effort and experience. Using an adaptation of the scale from Hutchinson & Tenenbaum (2006), participants were asked the amount of effort they spent on their activity from the previous session. Moreover, we asked participants about their experience with their activity through a free-text question. After describing their experience, participants could provide modifications in a second free-text response. Table S4 provides details on these three measures.

Barriers and motivators for doing the activities. We asked participants about their barriers and motivators for doing their assigned activities using two free-text questions (Table S4).

Views on interaction scenarios. Each interaction scenario ended with a question about whether participants would engage in the shown interaction if they were the persona from the video. Participants were asked to provide a rating on a scale from −5 to 5 and a free-text response after the prompt “Why do you think so?” Table S5 shows the question and scale endpoints for each interaction scenario.

User characteristics. We measured several user characteristics to explore their effect on the other measures. This included quitter and non-smoker self-identity measured with 3 items each based on Meijer et al. (2016) and physical activity identity based on an adaptation of the Exercise Identity Scale by Anderson & Cychosz (1994) to physical activity. All identity-related items were measured on five-point Likert scales. In addition, we measured the Transtheoretical Model (TTM)-stage for becoming physically active based on an adaptation of the question by Norman et al. (1998) to physical activity, and people’s Big-Five personality based on the 10-item questionnaire by Gosling, Rentfrow & Swann, 2003. The 10-item questionnaire by Gosling, Rentfrow & Swann (2003) was chosen due to its brevity and use in previous work on individual differences in behavior (e.g., Kaptein & Eckles (2012)). Despite its brevity, its convergent correlations compared to longer questionnaires such as the 44-item Big-Five Inventory (see John & Srivastava (1999)) have been found to be substantial (Gosling, Rentfrow & Swann, 2003). We also gathered information from participants’ Prolific profiles. This included their age range (e.g., 21–25), smoking frequency, weekly exercise amount, household size, and their highest completed education level. We used the education level as a measure of socioeconomic status, as is commonly done in smoking research (Meijer et al., 2016).

Participants

Eligible participants were those who were fluent in English, smoked tobacco products at least once per day, were contemplating or preparing to quit smoking (DiClemente et al., 1991), were not part of another intervention to quit smoking, and provided informed consent. Further, we aimed to increase the quality of the responses by requiring participants to have at least one completed study and an approval rate of at least 90% for their completed studies on Prolific. A total of 1,406 participants started the prescreening questionnaire, and 485 of the 922 eligible participants successfully responded to both interaction scenarios in the post-questionnaire. Participants had about 1 day to respond to their invitation to the pre-questionnaire, 3 days for the sessions, and 7 days for the post-questionnaire. The participant flow is presented in Fig. 1.

Participants who successfully completed a study component were paid based on the minimum payment rules on Prolific (5 pound sterling per hour). Since some participants faced difficulties accessing the videos of the interaction scenarios, participants who completed everything but part of the scenario questions in the post-questionnaire were also paid (N = 15). Participants were told that whether they did and how they reported on their assigned preparatory activities would not affect their payment. This was to account for self-interest and loss aversion biases. Self-interest bias can come into play when there are incentives that motivate participants to respond in a certain way; loss aversion bias can arise when participants suspect that they may not get paid fairly and thus choose not to participate or to drop out (Draws et al., 2021).

Participants on Prolific were nationals of or lived in member countries of the Organisation for Economic Co-operation and Development (OECD) with the exception of Turkey, Lithuania, Colombia and Costa Rica and the addition of South Africa (Prolific Team, 2022). Of the 671 participants with at least one valid free-text response, 349 were female, 310 were male, and 12 indicated a different gender or provided no information. The youngest participant was 18 and the oldest 74. With regards to smoking behavior, participants could be characterized as smoking once (5.37%), 2–5 times (24.59%), 6–10 times (31.74%), 11–19 times (28.32%), or more than 20 times (9.54%) per day. Moreover, 78.69% of participants indicated having previously quit smoking for at least 24 hours. An overview of these and further participant characteristics is provided in Table S6.

While sample sizes are less relevant for Bayesian analyses like ours than for frequentist ones (Chechile, 2020), we conducted a power analysis to get an idea of the statistical power of the quantitative part of our analysis in which we compute Spearman correlation coefficients. Following the Monte Carlo approach described by Kruschke (2014), we used 1,000 simulations of two standardized variables with a medium correlation of 0.3 according to Cohen (1992). For each simulation, we computed the 95% Highest Density Interval (HDI) for the correlation, with an HDI being “the narrowest interval containing the specified probability mass” (McElreath, 2020). The power was then calculated as the fraction of simulations in which the lower bound of the HDI was greater than zero. The result was a power of 0.68 for a sample size of 71, a power of 0.95 for a sample size of 148, and a power of >0.99 for a sample size of 300. These sample sizes are the smallest, median and largest number of samples we obtained for a group of interaction scenarios used in our quantitative analysis.

Procedure

Participants meeting the qualification criteria, passing the prescreening, and successfully completing the pre-questionnaire were invited to the first of five sessions with the virtual coach Sam. Those participants who successfully completed all five sessions were invited to the post-questionnaire. The post-questionnaire first asked participants about their effort spent on and experience with their last activity, then asked them about their motivators and barriers for doing their activities, and finally showed them 2 interaction scenarios. Before each scenario, participants were told that they would see a video and asked to turn on their audio. Underneath the video, we provided a link to the video on YouTube in case participants could not see the video in Qualtrics. Once the duration of the video had passed, participants could proceed to the next page to provide a rating and a free-text response for the scenario. Invitations to the next session or post-questionnaire were sent about 2 days after completing the previous session. Showing the interaction scenarios after participants had interacted with the virtual coach in five sessions spread over at least 9 days ensured that participants had personal experience of interacting with a virtual coach. Using an operational system has been described as crucial to be able to see possible benefits of health information technology (Canada Health Infoway, 2013).

Data preparation and analysis strategies

Data preparation

We preprocessed the gathered data by (1) using only data from sessions and the post-questionnaire if participants passed at least one attention check in the respective component, (2) using the first recorded submission for a study component if participants did the component more than once, (3) removing ratings and free-text responses for the interaction scenarios for people who wrote in their free-text responses that they could not see the video (N = 2), and (4) anonymizing free-text responses by removing potentially identifying or sensitive information. In addition, we computed the reliability of the items corresponding to the quitter, non-smoker, and physical activity identity measures. Since the reliability was sufficiently high for quitter (Cronbach’s α = 0.76, N = 671), non-smoker (Cronbach’s α = 0.69), and physical activity identity (Cronbach’s α = 0.89), we used the means of the items as index measures. We also reversed the scale for the TTM-stage for becoming physically active such that a higher value denotes a higher stage of change.

Analysis

We took a mixed-methods approach and proceeded in four steps to analyze the data. These steps were the thematic analysis steps described by Braun & Clarke (2006) with the addition of triangulation based on literature and quantitative results. We used two types of triangulation: method and investigator triangulation (Nancy Carter, Bryant-Lukosius & Alba DiCenso, 2014). Method triangulation was performed using data on both people’s actual behavior from their activity experiences and efforts as well as their views on possible behaviors from their free-text responses and ratings for the interaction scenarios. We also used data on user characteristics (e.g., physical activity identity) and findings from the literature. Regarding investigator triangulation, two researchers with different backgrounds were involved in all parts of the analysis. The result are the analysis steps that we now describe in detail.

Preparation of coding scheme. To create our coding scheme, the first author (NA) with a background in artificial intelligence and eHealth first familiarized herself with the data by reading all free-text responses and noting initial inductive codes. These codes were further refined deductively by looking through literature on technology acceptance and use, human motivation and behavior, and perceptions of virtual agents and robots. This included the two versions of the Unified Theory of Acceptance and Use of Technology (UTAUT) (Venkatesh et al., 2003; Venkatesh, Thong & Xu, 2012) (including their extensions with autonomy (Lakhal, Khechine & Pascot, 2013; Khechine & Augier, 2019), self-efficacy (Hewitt et al., 2019), and characteristics of the technology, situation, task, individual and other humans (Brown, Dennis & Venkatesh, 2010)), self-determination theory (Deci & Ryan, 1985), the Capability-Opportunity-Motivation-Behavior (COM-B) model of behavior (Michie, Van Stralen & West, 2011), barriers to behavior (Alfaifi, Grasso & Tamma, 2018), the findings by de Graaf, Allouch & Van Dijk (2015) on users’ experiences with a social robot, and the Ability-Benevolence-Integrity model of trustworthiness (Mayer, Davis & Schoorman, 1995). A draft coding scheme was discussed with SV who has a background in interaction design and had also read responses and formulated initial codes. The final coding scheme consisted of three levels, with four codes at the highest level, 15 codes at the second level, and 86 codes at the third level. Codes thereby captured both semantic and latent meanings of the responses (Braun & Clarke, 2012). The coding scheme is shown in Fig. S7.

Coding of free-text responses. All free-text responses were coded by NA based on the developed coding scheme. Multiple codes were used if relevant. We assessed the reliability using double coding. The second coder SV was further trained by independently coding six sets of ten responses and discussing the coding with NA after each set. Then, SV coded 100 responses. These 100 responses were chosen randomly but such that there were at least six responses per question (i.e., the 13 interaction scenarios, barriers, motivators, and activity experiences). The number of double-coded responses was selected to allow for an error margin of at most 10% to be obtained when calculating percent agreement (Gwet, 2014). We obtained moderate agreement (Cohen’s κ = 0.41) (Landis & Koch, 1977) at the third coding level. Due to its more robust nature (Gwet, 2021), we also computed the Brennan-Prediger κ (Brennan & Prediger, 1981) for a value of 0.97. Since participants primarily corrected spelling and grammar errors in their modifications of their activity experience answers, these modifications were excluded from further analysis.

Triangulation with literature and quantitative results. As NA and the third author (KP), with a background in psychology, gained insights from the coded free-text responses, literature and quantitative results were used to triangulate the qualitative results. Relevant literature came from diverse research fields such as eHealth, behavior change theories, human-robot interaction, and various application domains. Moreover, we incorporated two types of quantitative data. First, we computed means and Bayesian credibility intervals for the ratings per interaction scenario. The credibility intervals we report are the 95% HDIs. Second, we computed Spearman correlation coefficients between user characteristics (e.g., physical activity identity) on the one hand and users’ activity efforts and ratings for groups of interaction scenarios on the other hand. Note that we combined scenarios about similar interactions into groups to facilitate their discussion, as shown in Table S2. We conducted Bayesian tests for the correlations using the Bayesian First Aid R-package (Bååth, 2014) and report the median values and 95% HDIs. We classified the size of the resulting correlations using the guidelines by Cohen (1992). Furthermore, we calculated the posterior probability that a positive correlation is greater than zero and evaluated the probability using the guidelines by Chechile (2020).

Search, review and definition of themes, and production of the report. NA and KP examined the results to identify overarching themes. A final set of themes was obtained using multiple rounds of discussion. To produce the report, which is the last thematic analysis step described by Braun & Clarke (2006), NA selected participant responses that illustrate the themes. Participants are referred to by numbers (e.g., P123).

Results

We depict the frequencies of the most frequent codes from our coding scheme in Fig. 2 and those of all codes in Fig. S9. Figure 3 further presents the ratings for the interaction scenarios. In addition, we show the correlations between participants’ activity efforts and ratings for the interaction scenario groups on the one hand and user characteristics on the other hand in Fig. 4. We will refer to these figures throughout our discussion of the themes. In this discussion, we move from the smallest unit of analysis, a behavior, to the user who performs a behavior, to another party that may be involved in a behavior, to the largest unit of analysis, which is the environment (Fig. 5). This approach follows the idea of distinguishing micro, meso, and macro elements of behavior (Jaspal, Carriere & Moghaddam, 2016) as similarly done in previous work (e.g., Schouten et al. (2017)).

Figure 2 Percentage of times that codes from the coding scheme appear in each response type as well as across all response types together.

We show only the percentages of those codes that appear in at least 4% of the responses for at least one response type. The response types are the activity experiences, barriers, motivators, and the groups of interaction scenarios.

Figure 3 Means and 95% HDIs for the intentions to engage in the interactions from the interaction scenarios.

Abbreviations: PA, Physical activity; HRS, High risk situation; SE, Social environment; SO, Significant other; GP, General practitioner.

Figure 4 Overview of Spearman correlation coefficients between participant characteristics on the one hand, and the effort participants spent on their activities and their intentions to engage in the interactions from the scenario groups on the other hand.

Value labels are provided for all coefficients with an absolute value of at least 0.2. The color scheme is based on the absolute values of the coefficients. Abbreviations: PA, Physical activity; TTM, Transtheoretical model; Exp., Experience; HRS, High risk situation; SE, Social environment; GP, General practitioner.

Figure 5 Overview of the main discussed themes for each of the four hierarchical units of analysis.

The four units are behavior, user (who performs a behavior), another party (that may be involved in a behavior), and environment.

Behavior

Perceived usefulness

The most frequent topics both overall and for the interaction scenarios and activity experiences revolved around the perceived usefulness of the behavior (Fig. 2).

Getting help, advice or tips and learning. Thinking that they would get help, advice, tips, or learn something by engaging in a behavior was the most frequent topic overall (13.97%) and for all interaction scenario groups except for the scenario about receiving motivational messages (Fig. 2). Participants’ concerns included whether the behavior would help to reach their goals (e.g., P283), teach them how to deal with cravings (e.g., P92), or serve as a prompt to reflect in general (e.g., P274) or on their current behavior (e.g., P507). Several participants also stated that they thought a behavior was (not) helpful without providing specific reasons for this evaluation (e.g., P224, P151). For example, some participants who were against involving their General Practitioner (GP) noted that they did not see any way in which their GP could help them (e.g., P345, P639): No i wouldn’t [consult my GP], i don’t think my GP could do anything to help. ( Consult GP in case of smoking relapse, P345)

Obtaining information or knowledge has previously been identified as a theme in participants’ thoughts on using a self-regulation-based eHealth intervention to increase physical activity and intake of fruit and vegetables (Poppe et al., 2017). It also plays a role in the context of eHealth applications for other domains, including self-management of chronic conditions such as chronic pain (Solem et al., 2019) and type 2 diabetes (Lie et al., 2017). The scoping review of Wilson et al. (2021) also showed that the opportunity to learn new information is a motivator for the use of eHealth tools by older adults. It has even been argued that gaining knowledge is such a crucial motivation for using online activities and applications that it makes users active consumers and producers of health knowledge (Lupiáñez-Villanueva, Ángel Mayer & Torrent, 2009).

Getting motivation or encouragement. One element participants were looking for in the behaviors was help in the form of motivation or encouragement, which was with 3.82% the second most frequent topic overall for “behavior” and with 40.85% the most frequent one for the interaction scenario about receiving motivational messages (Fig. 2). Concerns about receiving motivational messages included whether the messages would be tailored to the user and situation at hand (e.g., P212, P497), help to increase or maintain self-confidence (e.g., P158), or serve as a reminder of (reasons for) quitting smoking (e.g., P6, P25): I don’t generally respond much to motivational messages, but in this instance, anything that can serve as a reminder for why I want and need to do this so much, is a good thing. Being asked to reflect on our reasons for quitting is definitely a good thing and something I personally would benefit from. ( Receive motivational messages, P25)

Similar findings came to light regarding using a help button for High Risk Situations (HRSs). After pressing such a help button, the virtual coach would provide support for dealing with cravings or difficulties to do planned physical activity. Here 23.65% of responses referred to getting motivation or encouragement. Participants mentioned wanting to get motivation in general (e.g., P283, P617), be reminded of (reasons for) quitting smoking (e.g., P52, P291), or get the strength to resist cravings (e.g., P79). Some participants also explicitly mentioned the importance of being motivated by someone or something else (e.g., P57, P417, P636): … sometimes I need a bit more motivation than what’s going on in my head and need a little kick or nudge in the right direction ( Help button for PA HRSs, P636)

These findings coincide with work by Poppe et al. (2017), which found that the opportunity to be motivated by being reminded of one’s goals was a reason for participants to prefer a mobile application instead of a website for increasing physical activity and the intake of fruit and vegetables. Moreover, Kulhánek et al. (2018) found motivation strengthening to be a frequently mentioned benefit of a conversational agent that assists with quitting smoking.

Improving feelings or mood, novelty, and comparison with others. Besides getting help and motivation, several other topics related to the perceived usefulness of the behavior. This included, for instance, whether engaging in the behavior improved one’s feelings or mood (e.g., P50, P244), whether the behavior was novel (e.g., P236, P464), or whether the behavior made a comparison with others possible (e.g., P310). Giving users the option to compare their performance to others, for example, can be an effective motivation strategy in persuasive games (Orji, Vassileva & Mandryk, 2014), just as novelty can be motivating (Croes & Antheunis, 2021; Sadeghi et al., 2021). According to this novelty effect, people are initially curious about a new system or technology and have high expectations regarding its usefulness. Yet, these perceptions decrease over time as the system’s limitations become apparent. The novelty effect was implied by one participant who gave the following response when asked to describe their motivators for doing their preparatory activities: curiosity at first, but that waned ( Motivators, P338)

Recommendations for addressing the perceived usefulness of the behavior. The central role perceived usefulness plays for the acceptance of a system has been modeled by the UTAUT (Venkatesh et al., 2003), which posits that the extent to which a person thinks that using a system will lead to personal performance improvement influences the intention to use that system. This effect on behavioral intention has been illustrated in studies of technologies in diverse contexts such as an app for insomnia treatment (Fitrianie et al., 2021b) and a socially assistive robot (Fridin & Belokopytov, 2014). One approach to increase the perceived usefulness is to tailor advice or content in general to users’ preferences. Previous work has, for example, attempted to automatically optimize suggested activities for older adults (Costa et al., 2016), physical activities for adults (Rabbi et al., 2015), or breast cancer screening recommendations (Ayer et al., 2016). Yet, future work on such automated approaches should take the novelty effect into account. An alternative is to use a participatory approach in which potential users can contribute to the design (Davis et al., 2021). Yet, our participants often drew conclusions about the usefulness of an intervention component without having tried it. This suggests that other elements such as telling people that the content is meant specifically for them (Dijkstra, 2016) or explaining in more detail why and how something helps them (Horsch et al., 2015) may be necessary.

Lastly, since getting motivation or encouragement was frequently sought out in our study, special attention should be given to how participants can be motivated or encouraged. Receiving motivational messages was overall seen positively in our study (Fig. 3), but we saw earlier that several people were concerned about the format of these messages. In addition, participants were also looking for motivation in other interactions, such as a discussion with their virtual coach about their repeated failure to reach their physical activity goals (Fig. 2).

Timing and intensity

The timing of the behavior is interesting in that it appears to play a role primarily for one type of interaction, namely, regular planning and reflection for HRSs. There appeared to be no consensus regarding an ideal time for this. For instance, some participants liked the proposed day (e.g., P470, P487) for physical activity- or time of the day (e.g., P121, P493, P627) for smoking-related planning and reflection. But others found the suggested timing for planning and reflection to be inappropriate: Maybe [I would make a plan with my virtual coach] on a different day, e.g friday for the week ahead starting on Monday. I do like the planning of the week and think ahead of the hurdles to avoid them. ( Plan for PA HRSs on Sundays, P262)

I think it’s great idea [to make a plan with my virtual coach,] just not sure if I would have enough time in the morning to do so (I start work at 6.30 a.m. so that would have to be quick plan) ( Plan for smoking HRSs in the mornings, P113)

Besides timing, participants were also concerned about the frequency (e.g., P158, P178) and duration (e.g., P362, P552) of (potential) application components.

Finally, while some participants favored the regularity of the planning and reflection interactions (e.g., P462), others expressed their preference for on-demand rather than regular support: I think consulting the virtual coach might be helpful – but the option to let the app know whenever I have a craving would be more convenient. ( Reflect on smoking HRSs in the evenings, P100).

Recommendations for addressing the timing and intensity of the behavior. Previous work on needs, barriers, and facilitators for older adults to use eHealth applications found instant availability of help through such applications to be a facilitator (Vergouw et al., 2020). However, given the diversity of user needs and preferences, there appears to be a need for tailored timing and intensity of intervention components. The importance of personalizing these elements of health behavior change applications has previously been pointed out by Dijkstra (2016). Approaches to achieving such personalization include both letting participants choose themselves and automatically determining opportune moments (Horsch et al., 2017). Letting participants choose acknowledges that users may know best what time suits them and that supporting autonomy and competence may increase motivation and performance (Ryan & Deci, 2000). In contrast, determining opportune moments for the user aims to account for the fact that people’s self-knowledge is generally not very accurate (Vazire & Carlson, 2010; Dunning, 2012). A middle ground may be to support users in systematically finding out what suits them (Karkar et al., 2016).

User

Importance of and motivation to change

Importance of change. When it comes to physical activity, several participants pointed out that becoming (more) physically active was not important to them. Reasons included already being physically active (e.g., P143, P533) and not seeing how physical activity helps to quit smoking (e.g., P657). Some participants, for instance, thought that working simultaneously on becoming physically active and quitting smoking was too difficult: It’s a bit condescending. Giving up smoking is hard enough without having to do a fitness regime also. ( Follow PA program while quitting smoking, P464).

Notably, this was even though participants were informed about the potential positive effects of physical activity on quitting smoking at the start of the study as well as every time they were assigned an activity for physical activity increase. Participants who were in favor of physical activity-related behaviors, on the other hand, frequently pointed out the benefits of physical activity both in general (e.g., P245, P335) and for quitting smoking specifically: I really think that physical activity could help me to quit smoking forever ( Motivators, P543)

The TTM provides a framework for assessing participants’ views on the importance of changing a behavior. Stage one, the precontemplation stage, is one in which people do not aim to change their behavior within the next six months and are often un- or under-informed about the consequences of their behavior (Prochaska, Redding & Evers, 2015). As people progress through the stages, they become increasingly aware of the consequences of their behavior(s) and ready to take action. The relationship between TTM-stage and taking action was observable in our study. More precisely, we observed a small correlation of 0.21 between people’s TTM-stage for becoming physically active and the effort they spent on their activities (Table 1). Hence, participants in higher stages overall spent somewhat more effort on their activities. Notably, we find this association between the TTM-stage for becoming physically active and the effort spent on activities even though only half of the activities were targeted at physical activity.

Table 1 Results of Bayesian analyses of Spearman correlations between user characteristics on the one hand and activity efforts and intentions to engage in the interactions from the scenario groups on the other hand.

For all seven correlations, at least 99.8% of the posterior distribution for the mean correlation was greater than zero. This leads to at least a very strong bet that the mean correlations are greater than zero (Chechile, 2020).

User characteristic	Effort/Scenario group rating	Median [HDI]	
Conscientiousness	Effort	0.20 [0.17, 0.24]	
Extraversion	Rating for ”Involve SE”	0.31 [0.15, 0.45]	
	Rating for ”Involve GP”	0.24 [0.08, 0.38]	
Household size	Rating for ”Involve SE”	0.26 [0.10, 0.40]	
PA identity	Effort	0.25 [0.21, 0.28]	
Smoking frequency	Rating for ”Discuss repeated failure PA”	0.41 [0.20, 0.59]	
TTM-stage PA	Effort	0.21 [0.17, 0.24]	
Note:

HDI, Highest density interval; SE, Social environment; GP, General practitioner; PA, Physical activity; TTM, Transtheoretical model.

Motivation to change. Once participants have become aware of the importance of changing their behavior, they also need to be motivated to do so. In our study, people’s motivation to reach their goals was with 50.00% the most commonly reported motivator for completing their activities, making it the overall most frequently mentioned topic for the “user” unit (Fig. 2). Goals which motivated participants included quitting smoking (e.g., P403, P463) and becoming more physically active (e.g., P126, P142), but also more general goals such as improving one’s health (e.g., P224, P390) and very individual ones such as being able to hold one’s breath for longer (P624). Sometimes, participants explicitly linked these other goals to quitting smoking or becoming more physically active: I was motivated by exploring all the positive health, family and psychological benefits that comes with quitting. And it fuels my drive to do the assigned activities. ( Motivators, P521)

my current health is not good so to try to improve this i need to give up smoking and do more exercise. This is my main motivation ( Motivators, P493)

At the same time, not wanting to quit smoking was mentioned by some participants as a barrier to doing their activities (e.g., P455, P591). Moreover, several participants were not confident in their ability to reach their goals (e.g., P111). These findings resonate with ones by Milcarz et al. (2019), according to which difficulties in quitting smoking and a lack of willingness to quit were the most commonly mentioned barriers.

Besides conscious or reflective motivation, automatic motivation also plays a role. Automatic motivation consists of automatic processes involving, for example, emotional reactions and reflex responses such as feeling excited at the prospect of going running in the evening (Michie et al., 2014). One way to capture people’s automatic motivation is to look at their self-identity (Michie, Van Stralen & West, 2011; West, 2013), which has been shown to predict people’s (intention and motivation to perform a) behavior (Priebe et al., 2020; Wolstenholme et al., 2021). In our results, we find some influence of self-identity on behavior based on a small correlation of 0.25 between physical activity identity and the effort people spent on their activities (Table 1). Thus, it appears that participants with stronger physical activity identities spent more effort on their activities. In addition, physical activity identity was also positively related to people’s willingness to follow a physical activity program besides quitting smoking, receive motivational messages, and involve their GP in the quit smoking process (Fig. 4). This is interesting, as some of these behaviors do not involve physical activity. One explanation could be that people with stronger physical activity identities felt more involved in the intervention as a whole due to the combination of quitting smoking and becoming physically active. One participant, for instance, pointed out the importance of being willing to become more physically active to be a suitable participant for the intervention: I’m fairly positive about this question because considering some of the activities were physical it would be important that the person was willing to be more physically active. ( Follow PA program while quitting smoking, P204)

This suggests that participants may need to be separately motivated to reach all goals an intervention puts forward. Notably, the association between quitter or non-smoker self-identities and the effort participants spent on their activities is much weaker than in the case of physical activity identity (Fig. 4). This likely is the case because participants had to be in either the contemplation or the preparation stage of the TTM for quitting smoking to be eligible for the study. As such, their identity as quitter or non-smoker may have been quite weak because not (yet) fitting with their (readiness to change their) smoking behaviors.

Recommendations for addressing the perceived importance of change and motivation to change. Our results show that as a first step, people need to be able to link a behavior to a desired outcome. For example, in this study, the benefits of physical activity for quitting smoking may have needed to be clearer. Our findings resonate with ones by Poppe et al. (2017), who showed that participants did not always agree with provided information on the positive effects of physical activity and fruit and vegetable intake. Participants who are not aware of the consequences of their behavior are often described as being in the precontemplation stage of the TTM. Important processes of change in this stage include receiving information about the behavior, evaluating how the problem affects one’s environment, and experiencing emotions about one’s situation and problems (Prochaska, Redding & Evers, 2015). Once people have learned about the consequences of their behavior, a next process of change may be self re-evaluation or realizing that the behavior change matches one’s identity better (Prochaska, Redding & Evers, 2015). Several participants of our study mentioned, for example, that an activity asking them to think about their feared or desired future self concerning quitting smoking or becoming more physically active was a major motivator (e.g., P243, P263): The task where I was to think about what I would be in the future gave me a huge boost for the rest of the tasks ( Motivators, P592)

Thinking about oneself in this way may weaken identity related to a problem behavior and strengthen identity related to one’s non-problem behavior (i.e., non-smoking if the problem behavior is smoking). Consequently, as identity represents a strong form of automatic motivation, such a change in identity can lead to successfully changing (health) behaviors such as smoking (Williams et al., 2002; Park et al., 2021). In addition, people need to gain confidence to change their behavior. Ways to increase confidence include allowing users to make small wins (Amabile & Kramer, 2011), letting them observe a relatable other succeed (Bandura, 1977), verbally persuading them (Strecher et al., 1986), or improving their mood (Kavanagh & Bower, 1985).

Autonomy

A topic that appeared several times in participants’ responses was autonomy (Fig. 2). For example, one participant perceived asking for help in a physical activity HRS as being a sign of excessive dependence on others, especially since they regarded the situation as not serious enough to require help: If I needed help like this just because I wasn’t exercising one evening I would feel very concerned that I could no longer stand on my own two feet in life. This is going too far. We are adults and this kind of hand holding is not healthy or helpful. ( Help button for PA HRSs, P133)

While a need for autonomy was put forward in several different interactions, the percentage of times this topic appeared was with 8.39% notably high in the context of involving one’s GP (Fig. 2). Participants mentioned wanting to decide for themselves whether to involve their GP instead of following the advice of the virtual coach (e.g., P429, P528). They also mentioned wanting to keep trying to change their behavior themselves (e.g., P472, P660), or only contacting their GP in severe cases (e.g., P85, P539). One participant also emphasized that seeing a GP is not helpful unless one is also motivated to quit smoking: … I believe that quitting smoking comes entirely from self motivation. Even if I would go to my GP, if I’m not personally committed it would be just a waste of time and money ( Consult GP at start of quit attempt, P624)

According to self-determination theory, people need to feel in control of their behaviors and goals to initiate behavior (Ryan & Deci, 2000). As such, the proposal to involve a GP likely violated this need. Besides our study, other eHealth interventions have identified autonomy and independence as needs for long-term weight maintenance (Asbjørnsen et al., 2020), healthy living for cardiovascular disease prevention and rehabilitation (Breeman et al., 2021), and self-management of chronic pain (Solem et al., 2019). The latter study also explicitly reported the difficulty of finding a balance between asking for help and being independent. Interestingly, however, our participants’ overall stance on involving the GP was also much more negative than for the other proposed interactions, with the mean of the credible interval even being less than zero for contacting the GP at the start of the quitting process (Fig. 3). This contradicts findings of previous work, which show that smokers tend to accept unsought conversations about smoking with their GP (Lems, 2006). Therefore, it was likely not the involvement of a GP alone that concerned participants in our study, but rather the way and personal situation in which involving a GP was proposed.

Recommendations for addressing users’ autonomy. Our results suggest that when a virtual coach recommends help for participants, their need for autonomy can be violated. Thereby, it appears to be crucial to not only consider what is recommended but also how and when. Interesting work in this regard has been conducted by Tielman et al. (2019). Their model for referring patients to human care is based on a combination of willingness to see a human and severity of the situation. If the situation is not severe and a patient is not in favor of seeing a human, it may be better not to actively try to persuade a person to see a human. The reasoning behind this is based on social judgment theory (Sherif & Hovland, 1961), which posits that any recommendation made to a person who is against a suggestion will likely fail and make the user more opposed to the idea. Besides considering people’s willingness to contact a human and the severity of the situation, it may also be useful to use a different formulation for the recommendations. For example, more emphasis could be placed on persuading people using, for instance, testimonials (Dijkstra, 2016). Alternatively, one could explain how the recommendation is in line with a user’s values (Dijkstra, 2016), or formulate utterances more carefully, as suggestions and less as commands (Free et al., 2009).

Personal characteristics

Besides people’s self-identity and TTM-stages, we also looked at the effects that their smoking frequency, weekly exercise amount, age, education level, or personality may have (Fig. 4). While previous work suggests a relationship between socioeconomic status and smoking (Fernández et al., 2006, Hiscock et al., 2012) and physical activity behaviors (Scholes & Neave, 2017), as well as between age and smoking behaviors (Kviz et al., 1994; Li, Hsia & Yang, 2011), our results do not suggest a strong effect of education level and age on people’s activity efforts and ratings of the interaction scenarios (Fig. 4). What we did observe is a moderate correlation of 0.41 between smoking frequency and willingness to discuss repeated failures to reach physical activity goals (Table 1). One explanation for this observation could be that as heavier smokers are more likely to seek help in quitting smoking (Chaiton et al., 2007), they may be more open to receiving support. In addition, we find several small to moderate correlations between people’s Big-Five personality dimensions and their activity efforts and interaction scenario ratings (Fig. 4). For example, there is a small correlation of 0.20 between conscientiousness and activity effort (Table 1). This matches the observation that the code “conscientiousness” appeared several times in relation to the preparatory activities (e.g., P320, P380): I approach the activities with clear focus and intense dedication and discipline ( Activity experience, P521)

Moreover, extraversion correlated with people’s willingness to involve both their social environment and GP as shown in Table 1. This is in line with the observation that, especially with regards to involving their social environment, several participants expressed their desire for privacy (Fig. 2). While some participants wanted privacy in general (e.g., P495, P573), others were concerned explicitly with not wanting to be seen failing: I would prefer to keep it to myself until im confident that i can kick this habit for good ( Tell SE about quit attempt, P111)

… I have done that [i.e., tell my social environment about my quit attempt] before and felt pressure to stop. Then you feel like a failure if you dont succeed ( Tell SE about quit attempt, P79)

Overall, however, the effects of personality were relatively small. This is in line with previous work, which found primarily small effects of personality on physical activity (Rhodes & Smith, 2006; Rhodes & Boudreau, 2017) and smoking behaviors (Terracciano & Costa, 2004; Munafo, Zetteler & Clark, 2007).

Recommendations for addressing users’ personal characteristics. Our results are indicative of personal characteristics playing a role in people’s preferences for using an eHealth application. The consequence is that it may be harder to convince some participants of the merits of certain behaviors, such as involving their social environment. However, our observed effects were primarily small, and the topics “want for privacy” and “conscientiousness” appeared relatively infrequently in the free-text responses (Fig. S9). Therefore, rather than directly tailoring application components to people’s characteristics, the most straightforward approach may be to simply acknowledge that people differ in their preferences and leave room for these differences. Interestingly, we find that, even when not explicitly asked to, people tend to follow their preferences. For example, several people modified a preparatory activity in which they were asked to visualize smoking or becoming more physically active as a fighting match. Instead of a fighting match, they imagined a bike race (P104), a soccer match (P539), or a verbal fight (P87). While not all people provided reasons for doing so, P539, for instance, mentioned imagining a soccer match because of being a soccer fan.

Other party

Now we zoom out one more unit to the other party involved in an interaction. This includes the virtual coach, social environment, and the GP, as well as people featured in educational videos that were part of some preparatory activities.

Companionableness. Appearing in 1.82% of the 4,839 responses, companionableness was the most frequently mentioned topic relating to other parties involved in a behavior (Fig. 2). To participants, this included having another person or friend to talk to (e.g., P260, P645), not feeling alone (e.g., P442, P513), finding the other party condescending or patronizing (e.g., P274, P423), feeling close or connected to the other party (e.g., P89, P95, P476), being able to relate to the other party (e.g., P68), being able to share accomplishments (e.g., P528, P635), and feeling supported and not pressured (e.g., P416, P547). Support from others was frequently mentioned as a motivator for doing the activities (Fig. 2), and companionableness of the other party appeared to be important for support to be helpful: Having an extra support is really helpful and if it comes trough an important person it has more impact in my choices ( Discuss with an SO how they can support the quit attempt, P593).

Depends on what happens after I press the button. Will the ai try to act as my friend and scold me? ( Help button for smoking HRSs, P263)

Further support to this is given by the observation provided in Table 1 that participants with a larger household, and thus likely a larger and closer social environment, were more willing to involve their social environment.

Previous work has described the importance of companionableness, such as being able to count on a social robot (de Graaf, Allouch & Van Dijk, 2015), trust one’s primary healthcare providers as well as the received health-related information in the context of an eHealth application for cardiovascular disease and dementia (Akenine et al., 2020), and feel supported in eHealth applications for both long-term weight maintenance (Asbjørnsen et al., 2020) and knee osteoarthritis (Nelligan et al., 2020). The pilot study for the smoking cessation intervention txt2stop also found that people disliked messages that were seen as patronizing (Free et al., 2009). Similar recommendations were formulated by Michie et al. (2012) for the internet-based smoking cessation intervention StopAdvisor. Moreover, Henkemans et al. (2017) showed in the context of a robot playing a self-management education game with children with type 1 diabetes that the children answered more questions correctly and perceived the interactions as more pleasurable when the robot was designed to account for the children’s needs for, among others, relatedness.

Nature: Human vs AI. Importantly, (lack of) companionableness was ascribed to both the virtual coach and humans. However, the nature of the other party was referred to by some people. For example, some people were entirely against using a virtual coach: If it was an actual person, I could probably consult, but being a virtual coach, I would not be as interested. ( Reflect on PA HRSs on Sundays, P573)

I would never consult a virtual coach ( Reflect on smoking HRSs in the evenings, P330)

However, more commonly, people expected certain characteristics or abilities to (not) be present in a virtual coach compared to a human. One such characteristic was situational awareness, or the ability to understand and tailor to the individual user and their situation: … I’d love to see tips, but the reason why is much deeper and requires a human. ( Reflect on PA HRSs on Sundays, P452)

Other aspects mentioned by participants with regards to the nature of the other party included a lack of empathy (e.g., P630) and finding a virtual coach less motivating than a human: I don’t think I’d consult any virtual coach on this issue, as I find real people to be much more motivating. ( Discuss repeated failure of reaching PA goals, P416)

People who did like the virtual nature of the coach, on the other hand, mentioned fearing or not liking to contact real people (P222, P225) as well as feeling less embarrassed to admit failures to a virtual coach than to a human (P269).

Several of these factors have also been found to play a role in previous work. Issom et al. (2021), for instance, saw that the empathy expressed by a conversational agent was particularly appreciated, and de Graaf, Allouch & Van Dijk (2015) observed that the expression of human-like emotions by a social robot was perceived as important. Expressing empathy has also been shown to help a virtual agent to form and maintain a relationship with a user (Bickmore et al., 2005), which can support behavior change (Zhang et al., 2020). In addition, Issom et al. (2021) observed that the anonymous nature of conversing with a conversational agent was valued. Regarding situational awareness, both de Graaf, Allouch & Van Dijk (2015) and Issom et al. (2021) obtained similar findings in that participants preferred the social robot to understand more than just pre-programmed commands and that users of conversational agents requested more flexible answer choices. Notably, there can also be differences between AI embodiments. For instance, Sinoo et al. (2018) saw that children’s feelings of friendship were stronger toward a physical robot than an avatar.

Accountability. The fourth most frequent topic with regards to the “other party” unit, after companionableness, nature, and situational awareness, was with 0.60% accountability (Fig. 2). People felt accountable to the virtual coach (e.g., P31, P466) and their social environment (e.g., P475, P638), although accountability was perceived as stronger when coming from humans: With my experience with Sam I realise that a virtual assistant can really help and I also think about the fact that if I fail at some point it’s “more ok” to let down a fake person than someone real that I care about a lot. ( Reflect on smoking HRSs in the evenings, P593)

As a result, accountability was sometimes seen as too strong when coming from the social environment and too weak when coming from the virtual coach: Adding peer-pressure to an already stressful situation would not be useful ( Tell SE about quit attempt, P264)

It’s easy to dismiss a virtual coach, maybe it works for people who are very committed to quit and would be a reminder ( Plan for smoking HRSs in the mornings, P273)

The importance of feeling accountable to somebody comes forward in work by Nelligan et al. (2020), who found accountability to be part of the primary themes describing attitudes and experiences in the context of an eHealth application. The relevance of accountability was also observed by Lie et al. (2017), who saw that individuals with type 2 diabetes felt more accountable to regular health consultations than virtual ones.

Recommendations for the other party. Our results show that companionableness is a key ingredient in interactions with another party. While addressing the perceived companionableness of the social environment or GP may lie outside the reach of an eHealth application, improving the one of a virtual coach does not. As a start, it is important to be aware that it is not only possible for people to form a relationship with a machine or computer (Croes & Antheunis, 2021), but that people also tend to treat their computers as social beings (Nass & Moon, 2000). Relatively simple strategies can help to improve such a relationship with a system. This includes giving a virtual coach a name to increase its social presence (Zhang et al., 2020), avoiding repetitiveness and predictability to improve engagement, enjoyment of the interactions, and motivation to perform an advocated behavior (Bickmore et al., 2005; Croes & Antheunis, 2021), and trying to avoid responses that may be seen as too enthusiastic (Free et al., 2009). Other aspects such as learning from individual conversations, building on and referring to previous conversations, and conveying in-depth information on various topics as humans commonly do, however, remain open challenges (Croes & Antheunis, 2021). Nevertheless, paying close attention to the relationship between a user and a virtual coach is likely to pay off, as a good relationship can support behavior change (Zhang et al., 2020). Notably, however, improving the relationship between a user and a virtual coach should not come at the expense of transparency: the user needs to be aware that they are interacting with a virtual coach and not a human (Madiega, 2019).

Environment

Difficulty of integrating (health) behaviors into people’s busy lives

People are busy with other things. The most frequently mentioned topic with regards to the environment was with 5.04% whether participants had enough time to perform a behavior, and especially to complete their preparatory activities (Fig. 2). Notably, 18.80% of barriers to completing the activities involved the availability of time, making it the most commonly mentioned barrier (Fig. 2). It turned out that participants tended to be busy with their daily lives, including work, child care, and daily chores, and that these tasks left no time (e.g., P111, P432), caused people to be too tired (e.g., P66, P140, P262), or made it difficult for people to focus on their preparatory activities (e.g., P495, P600, P642): I run out of time with home life taking a focus so didnt get time to complete ( Activity experience, P432)

It was hard to plan the exercise in my daily planning, as the days are full and i’m very tired at the end of the day ( Activity experience, P262)

I thought about it [i.e., the person I would like to be once I have successfully quit smoking] for a couple minutes with the intention to write it down, but got distracted with other things. It did remind me that I did want to quit though. ( Activity experience, P642)

When participants had no time to do their activities, they reported having spent their time on other priorities such as home life in the example of P432. A similar phenomenon was found in Lie et al. (2017) where people dropped out from an eHealth intervention for self-management of type 2 diabetes because the daily life took the front stage. Similarly, a scoping review by Wilson et al. (2021) saw that the inability to incorporate a behavior into one’s routine was a barrier to using eHealth tools. In addition, the statement by P262 (i.e., being too tired to do a behavior) suggests that the participant expected the behavior to require a considerable amount of effort, which was also one of our codes for the “behavior” unit (Fig. 2). The expectation of effort or effort expectancy is one of the predictors of the intention to perform a behavior according to the UTAUT (Venkatesh et al., 2003), and several studies have confirmed this relationship in the context of eHealth tools (e.g., Boontarig et al. (2012), Quaosar, Hoque & Bao (2018), Fitrianie et al. (2021b)).

Prompts and triggers are helpful. The second most frequent topic for “environment” was with 3.49% of all responses whether participants had prompts or triggers for doing a behavior, and especially to complete a preparatory activity (Fig. 2). Participants commonly reported that they forgot to do (part of) their preparatory activity (e.g., P269, P331). Reasons included being busy (e.g., P43, P527), and that some activities (e.g., tracking one’s smoking behavior) required one to remember to do something at specific times (e.g., P182, P227): I approached the activity with a positive thought but found myself forgetting to record the timea i had a cigarette. Thia was mainly due to smoking when i had opportunity for a quick break so was always rushing. ( Activity experience, P182)

Participants also commonly started an activity but then stopped to do something else and forgot to get back to it: I watched 1 minute but I started different activity and forgotten about it. ( Activity experience, P281)

The importance of prompts and triggers is further illustrated by the fact that several participants who did complete their activities reported making use of them. Participants mentioned completing their activity right after the session had ended (e.g., P186, P480), once they received the reminder message we sent after the session on Prolific (e.g., P180, P393, P417), or based on a reminder they had set themselves (e.g., P7, P346): I set a remind on my smartphone to recall me the activity, so yesterday, in my bed before sleeping, I thought about who I want to be once I have quit smoking ( Activity experience, P346)

A scoping review by Wilson et al. (2021) found a lack of reminders or alerts to be a barrier for older individuals to use eHealth tools and their presence to be a motivator. Similarly, Horsch et al. (2015) saw that people favored the use of reminders to help with forgetting in the context of insomnia treatment. Participants of the study by Horsch et al. (2015) also emphasized, however, that users themselves should set reminders. This links to the earlier discussed topic of autonomy in that users want to be in control of application components.

Recommendation for addressing the difficulty of integrating (health) behaviors into people’s busy lives. According to the COM-B model of behavior (Michie, Van Stralen & West, 2011), one predictor of behavior is whether people have the opportunity to perform the behavior. This includes sufficient time and prompts or triggers to remind them. Our results suggest that both of these factors tend to be lacking. A straightforward way to help people who lack time is to create action plans. Action plans specify where, when, and how one plans to do something to create a link between a cue and a behavioral response (Hagger & Luszczynska, 2014). Action plans have been effective in changing behaviors such as physical activity, smoking, and alcohol consumption (Sniehotta et al., 2005a; Hagger & Luszczynska, 2014). To further account for sudden barriers to doing a behavior such as being tired, one can specify coping plans to create a link between a possible risk situation and a feasible way of coping with it (Sniehotta et al., 2005b).

Another strategy could be to prompt participants to create reminders themselves or determine suitable times for sending automatic reminders (Horsch et al., 2017). This was actually recommended by one participant ( Activity experience, P247). It turns out that reminders are already one of the most common persuasive components of eHealth applications (Lentferink et al., 2017). However, it is important to keep in mind that a high effort expectancy and other more relevant priorities likely also play a role for somebody who is too tired or has no time to do a behavior. This shows that characteristics of the environment can be intertwined with ones of the behavior and the user.

Lastly, the topics “having enough time” and “having prompts or triggers” primarily appeared in participants’ statements about their actual behavior (i.e., their activity experiences and barriers) rather than their views on possible behaviors in the form of the interaction scenarios (Fig. 2). Thus, these factors are less evident to people when they are just asked about their views on possible behaviors. One likely explanation for this is the optimism bias, according to which people tend to be overly optimistic about themselves and their future (Weinstein, 1980). For example, the study of Horsch et al. (2015) showed that people tended to be rather optimistic about their future adherence to an eHealth application for insomnia treatment. Reasons may include relying too much on future willpower and ignoring things that could go wrong (Horsch et al., 2015). This underlines the importance of having participants interact with a system to get a thorough assessment of their needs (Canada Health Infoway, 2013). On the other hand, other topics, namely ones related to the other party, primarily appeared in the views on the interaction scenarios rather than statements about actual behavior (Fig. 2). Thus, combining data on actual and potential behaviors offers a clear benefit.

Helpfulness of support from social environment

We have already touched upon the role of the social environment in the context of characteristics of the other party that influence people’s views on interaction scenarios. However, our results also suggest the general importance of support from one’s social environment. In fact, 6.60% of participants reported support from their social environment to be a motivator for doing their preparatory activities (Fig. 2). The social environment supported these participants in their wish to reach their goals such as quitting smoking (e.g., P212, P475) and helped them to complete the activities: Wrote down on a list last night and discussed with partner. Helpful and motivating. ( Activity experience, P133)

An important form of support is not just verbal but also behavioral. Some participants felt less motivated because their social environment did not live according to their own behavioral goal of not smoking (e.g., P127, P207, P315), or because they did not feel part of a group that worked toward the same behavioral goal (e.g., P262): Probably the people watching the ad will think about the fact that usually smokers need motivation from other people, but often the motivation is not there because smokers tend to surround themselves with other smokers. ( Receive motivational messages, P207)

… As well as the motivation of other participants. When feeling you are a part of a group that want to achieve the same thing i feel that this is more motivated. ( Barriers, P262)

Previous work by Willemsen et al. (1996) found social pressure from the social environment of Dutch employees to be a predictor of intention to quit smoking, and Breeman et al. (2021) concluded that involving the social environment was a desired core attribute for an eHealth application for healthy living. In addition, Meijer et al. (2016) note that the support and social norms present in social environments can shape identities, which in turn can affect behavior as discussed previously. For example, according to the social identity model of recovery (Best et al., 2016), a person’s recovery identity in the context of addiction can become stronger if it is shared with other people who favor recovery.

Recommendations to address the helpfulness of support from one’s social environment. We find that support from one’s social environment can be motivating and can help to perform activities that are part of a behavior change intervention. A straightforward way to include social support in an eHealth intervention is to prompt participants to either tell their social environment about their behavior change process or discuss with a significant other about how they can support it. Both of these elements were generally seen positively by our participants (Fig. 3). However, it is important to keep in mind the earlier discussed personal characteristics such as a want for privacy and characteristics of the social environment that may influence whether people want to involve their social environment. Taking a similar approach to Tielman et al. (2019) and taking people’s willingness to involve a human and situation severity into account may be beneficial. Another approach may be to connect people who work toward the same behavioral goal. Promising results can be obtained with relatively simple solutions such as a WhatsApp group (Simons, van den Heuvel & Jonker, 2018). Such a group has the advantage that it is easy to implement and accessible due to the omnipresence of WhatsApp (Simons, van den Heuvel & Jonker, 2018).

Diversity of other environmental factors

Several other environmental factors such as not having access to the Internet on certain days (P233), not using one’s phone when at home or on specific days (e.g., P376, P624), not being able to access the videos contained in the activities due to one’s location (P668), poor weather (P468), or restrictions related to the COVID pandemic (e.g., P4, P631) were mentioned. Given the diversity of environmental factors that can play a role, it is likely difficult, if not infeasible, to specifically address all of them. Since such other environmental factors appeared relatively infrequently (Fig. S9), the most important insight may thus be to design an eHealth application in such a way that it leaves room for individual barriers and gives users resources to try to cope with these barriers themselves.

Discussion

Through a thematic analysis based on qualitative data, quantitative data, and literature, we have discussed 14 main themes that are associated with people’s actual behavior and views on potential interactions in the context of a virtual coach for quitting smoking and becoming more physically active. These 14 themes can be structured by assigning them to four hierarchical units of analysis. These units are “behavior,” “user” (who performs a behavior), “other party” (involved in a behavior), and “environment.” While these units provide a convenient frame for analysis, it is important to note that the observed themes often span multiple units or depend on themes in other units. For instance, the environment, user, and behavior are involved in observing that people are often too tired from their busy daily lives to perform a behavior. This is in line with previous work that has highlighted the interdependence of factors from the environment, user, and the technology (van Gemert-Pijnen et al., 2011).

Aligning time, perceived usefulness, and users’ goals. The most common topics for the “environment” unit were having enough time and prompts or triggers for doing something. One could address this by making suggestions at convenient times (e.g., Horsch et al. (2017)) or helping people create action plans (Hagger & Luszczynska, 2014). Yet, whether people have time for something and remember to do it likely depends on how useful they find it. For instance, some participants set their own reminders for doing preparatory activities. Those people likely perceived the activities as so useful that they wanted to ensure they did them. Perceived usefulness was the most common theme in participants’ responses, but it is also connected to another topic. More precisely, somebody who does not see the link between a behavior and their goals is likely to find the behavior less useful. Recall that the motivation to reach their goals was the most common motivator for people to do their preparatory activities. Those who mentioned this motivator likely saw how the preparatory activities aligned with their goals. Ways to strengthen the link between perceived usefulness and users’ goals include referring to people’s goals and beliefs when giving advice (Abdulrahman, Richards & Bilgin, 2021). Another approach, which motivated several participants, is to think about one’s feared or desired future self with regards to a behavior (Meijer et al., 2018; Penfornis, Gebhardt & Meijer, 2021). Given that these topics of having enough time, perceived usefulness, and users’ goals appear to be connected, addressing them together may be beneficial.

We have formulated recommendations for how to address each observed theme as part of our analysis. Besides these recommendations, we find that the following challenges would benefit from more attention in the future:

Is there a set of standard factors that are generalizable across domains? Our study was conducted in the context of smoking cessation and physical activity increase and with participants that were enrolled on the online crowdsourcing platform Prolific and hence had at least some experience with using digital services. Thus, the question regarding the generalizability of our findings to other types of behavior change, or even eHealth more generally, and users with less experience with digital services arises. In addition, as our participants were contemplating or preparing to quit smoking and were paid for completing the conversations with the virtual coach, it is not clear how our findings generalize to a setting with people who are not yet contemplating to change and do not receive such a payment. We think that there are two important steps. The first one involves carefully describing the study context to more easily compare it across studies. While progress has been made for systematically reporting behavior change techniques (Michie et al., 2013), others elements of the context such as the environment and virtual coach are often described less extensively and without clear guidelines. One useful direction in this regard is the work by Fitrianie et al. (2021a), which aims to create a questionnaire that reports characteristics of virtual agents in a standardized fashion. A second step is to examine which factors affect user needs and how these factors change as the study context varies. For instance, while several of our themes coincided with earlier studies on eHealth applications for other domains such as self-management of chronic conditions (Lie et al., 2017; Solem et al., 2019), more research is needed to determine how findings can be generalized. Other important characteristics of the study context include the embodiment of and way of interacting with an AI (e.g., Sinoo et al. (2018), Gupta, Hathwar & Vijayakumar (2020)) as well as whether the intervention is blended (e.g., Meijer et al. (2021)).

How to get input from (many) users? Some themes, such as lacking prompts and triggers or having no time to perform a behavior, primarily appeared in people’s descriptions of their actual behavior as opposed to their views on possible interaction scenarios presented in videos. This was the case even though people provided their views on the interaction scenarios after having experienced actual interactions in the form of doing suggested preparatory activities. The disparate theme distribution is likely due to the optimism bias, according to which people tend to be overly optimistic about their future (Weinstein, 1980). This underlines the need for carefully choosing a method and using multiple ones if a comprehensive understanding of users’ needs is desired. As done in our study, crowdsourcing can facilitate reaching a large and diverse number of people once one has chosen a method. Yet, it is very time-consuming to conduct a thematic analysis of many free-text responses manually. Thus, crowdsourcing needs to be supplemented with automatically extracting codes from text to allow large-scale thematic analyses to be adopted more widely. Promising results for identifying predefined codes exist in specific application areas such as cognitive therapy (Burger, Neerincx & Brinkman, 2021) or news (Odijk et al., 2013). However, as novel codes may appear in the data, such approaches need to be combined with ones for generating codes. First approaches exist (e.g., Liew et al. (2014), Leeson et al. (2019)), but challenges remain regarding explainability and trust, among others (Chen et al., 2018). While those challenges persist, automated approaches may be beneficial as an adjunct to qualitative analysis by informing the creation of codes, checking the accuracy of coding, or pointing out ambiguity (Chen et al., 2018; Leeson et al., 2019).

How to tailor application components? In several of our themes, there was an apparent need for tailoring application components to individuals or groups of users. For example, while some people favored involving their social environment, others expressed a need for privacy. However, it is not clear how such tailoring can best be accomplished. Options include letting users choose themselves to support autonomy and competence (Horsch et al., 2017), automatically tailoring to users to account for people’s lack of self-knowledge (Horsch et al., 2017), and helping users to self-experiment (Karkar et al., 2016). Moreover, each of these general approaches can be implemented in many ways, and they can also be combined. Ranjbartabar et al. (2021), for instance, employed users’ preferences as a starting point for subsequent automatic tailoring. Further research is needed to determine which approaches to tailoring are effective in increasing adherence and under which conditions. One promising way to test many tailoring approaches and configurations may be micro-randomized trials, which allow participants to be randomized hundreds of times during a single study (Klasnja et al., 2015).

Conclusions

In light of the dropout and lack of adherence common to eHealth application for behavior change, we need a better understanding of user needs and how to address them. We thus conducted a thematic analysis of people’s experiences with actual and views on potential behaviors. The context was a text-based virtual coach for quitting smoking and becoming more physically active.

We found that users’ needs are often interconnected and include characteristics of the behavior, the user, other parties such as the social environment, and the environment. We identified 14 main themes that describe users’ needs: of these, the perceived usefulness of behaviors is most prominent and relates to environmental characteristics such as having sufficient time and the user’s state such as their motivation to reach their goals. We publish our dataset with user characteristics and 5,074 free-text responses from 671 people to aid future work on understanding the interplay between users’ needs and characteristics. This dataset can also be used to improve preparatory activities for quitting smoking and becoming more physically active, as it contains 2,866 descriptions of experiences with 24 such activities.

Based on this analysis, we formulated recommendations for how users’ needs can be addressed in eHealth applications for behavior change. Besides the specific recommendations we provide for each need, we suggest that associated needs should be addressed together. Adherence could, for example, be strengthened by referring to users’ goals and their beliefs when giving advice on quitting smoking and increasing physical activity.

Supplemental Information

Supplemental Information 1 Preparatory activities.

Titles and formulations for the 24 preparatory activities that were used in the study, 12 each for smoking cessation and physical activity increase. 4 activities had another activity as prerequisite.

Click here for additional data file.

Supplemental Information 2 Interaction scenarios.

Some scenarios about similar interactions are grouped together to facilitate their analysis.

Click here for additional data file.

Supplemental Information 3 Links to the videos of the interaction scenarios.

We provide links to both the male and the female version.

Click here for additional data file.

Supplemental Information 4 Details on how we measured the activity feedback, and barriers and motivators for doing the preparatory activities.

Click here for additional data file.

Supplemental Information 5 Question and scale endpoints for each interaction scenario.

Click here for additional data file.

Supplemental Information 6 Participant characteristics.

Characteristics of the 671 participants with at least one valid free-text response.

Click here for additional data file.

Supplemental Information 7 Coding scheme.

The coding scheme consists of 4 codes at the highest level, 15 codes at the second level, and 86 codes at the third level. We show the reliability for each coding level.

Click here for additional data file.

Supplemental Information 8 Conversation example.

The example shows the start of the second conversational session with Sam, including responses given by a participant.

Click here for additional data file.

Supplemental Information 9 Percentage of times each code from the coding scheme appears in each response type as well as across all response types together.

The response types are the activity experiences, barriers, motivators, and the groups of interaction scenarios. N denotes the number of responses.

Click here for additional data file.

We would like to thank Dr. Eline Meijer for her feedback on the preparatory activities, Dr. Eline Meijer and Bouke Scheltinga for their input on the interaction scenarios, Shruthi Venkat for serving as a second coder, and Mitchell Kesteloo for helping to host the virtual coach on a server.

Additional Information and Declarations

Competing Interests

Author Contributions

Human Ethics

Data Availability

The authors declare that they have no competing interests.

Nele Albers conceived and designed the experiments, performed the experiments, analyzed the data, prepared figures and/or tables, authored or reviewed drafts of the article, and approved the final draft.

Mark A. Neerincx conceived and designed the experiments, authored or reviewed drafts of the article, and approved the final draft.

Kristell M. Penfornis analyzed the data, prepared figures and/or tables, authored or reviewed drafts of the article, and approved the final draft.

Willem-Paul Brinkman conceived and designed the experiments, authored or reviewed drafts of the article, and approved the final draft.

The following information was supplied relating to ethical approvals (i.e., approving body and any reference numbers):

The Human Research Ethics Committee of Delft University of Technology approved the study (Letter of Approval number: 1523).

The following information was supplied regarding data availability:

All data and analysis code are available at 4TU.ResearchData:

Albers, Nele; Neerincx, Mark A.; Penfornis, Kristell M.; Brinkman, Willem-Paul (2022): Users’ needs for a digital smoking cessation application and how to address them: Data and analysis code. 4TU.ResearchData. Dataset. DOI 10.4121/20284131.v2.

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
