# Peer review of "Users’ needs for a digital smoking cessation application and how to address them: A mixed-methods study"

_PeerJ, doi:10.7717/peerj.13824_

## Round 0.1 · original submission · Minor Revisions

Dear Authors,

I have now received the reviewers' comments on your manuscript. The reviewer have recommended publication, but also suggest some minor revisions to your manuscript.  Therefore, I invite you to respond to the reviewers' comments and revise your manuscript.

Sincerely,

Reviewer 1 ·

Basic reporting

The manuscript titled "Users' needs for a digital smoking cessation application and how to address them: A mixed-methods study" is written in a clear, professional English language. Theoretical background shows an adeguate state-of-the-art of published literature in the field. Structure of the manuscript conforms to PeerJ standards (but the number of characters in the abstract should be carefully checked), and figures are relevant, even if the content of some of them (e.g., Figure 2) is almost illegible. Raw data is supplied.

Experimental design

This is an original piece of research that falls within the scope of the journal. The research question is well defined, relevant and meaningful. This piece of research can contribute to fill a knowledge gap. Methodology is appropriate and accurate and follows high technical and ethical standards, but should be integrated with a priori calculation of statistical power.

Validity of the findings

Data are provided and clearly described for further investigations. Conclusions are clear and well stated, linked to the research questions and based on results.

Additional comments

Overall, this manuscript is well written and reports a well conducted piece of research. In terms of novelty of the content, the manuscript presents interesting preliminary evidence about the possible contribution of digital applications in supporting people to quit smoking. In terms of professional English (expression, grammar and spelling), the manuscript is fine. The journal "PeerJ" is an appropriate scientific venue for this manuscript. In terms of study design and methodology, both are appropriate. Lastly, in terms of ethical issues, I cannot see any significant problem.

However, I would suggest few minor changes that will improve the quality of the manuscript.

1) I think that tha abstract is a little bit longer than expected. I would recommend the authors to double-check the number of characters;

2) As previously mentioned, the content of the Figure 2 is almost illegible. I would recommend the authors to improve the clarity of this figure;

3) The research design would benefit from the inclusion of an a priori calculation of statistical power in order to demonstrate that the sample recruited was large enough to detect a significant effect at the significance level of α=0.05: as this study included a large sample of participants, the output of this procedure will probably be in favor of this study, making it more explicit its methodological strength;

4) The table 1 is interesting, but I would recommend the authors not to include it in the main manuscript but in the supplementary materials, as it is not crucial for the reader to get immediately this piece of information when looking at the main aspects of the study.

Reviewer 2 ·

Basic reporting

See below

Experimental design

See below

Validity of the findings

See below

Additional comments

This useful study aims to explore the reasons for reduced
compliance/adherence (and numerous drop-outs) reported in studies
conducted with the use of stop smoking smartphone applications.
It is generally well done, the problems considered are different and
the systematic study of the impact of the virtual coach as a function
of the various dimensions of the phenomenon seems to be well argued.
One of the major contributions, in my opinion, is the open-access
provision of the raw dataset. I have only some minor comments:
There is no info on usability and user experience of the cessation
app. This could be contrasted with the findings of the present study
(proof-of-concept) or can be discussed in the main text as a case study.
The study sample was made of Dutch users willing to participate in a
smoke quitting plan. The possibility of selection bias should be
addressed as a limitation.
From a technological point of view the authors are opting for a
standard approach (i.e. essentially test chats, and IA-based virtual
bots similar to Alexa, etc.). The possibility for alternative (i.e.
hybrid interaction) or more innovative approaches should be
acknowledged in the text.
Figure 1 is confusing. I suggest you to use a scheme similar to that
of CONSORT.
A direct link to “Sam” could facilitate direct access (table S1 is too
convolute to follow and should be reduced in length).
The questionnaires are well balanced, however a clear rationale for
the use of Big Five Questionnaire 10 items should be included.
It would have been helpful to provide a detailed characterization of
the smokers participating in the study. Have you included also people
that smoke two or three cigarettes day? And from how many years?
A limitation is that participants received a compensation and this
could cannot replicated in a naturalistic setting. This needs to
be addressed in the text.

---

## Round 0.2 · accepted · Accept

Many thanks for addressing all the issues.

Reviewer 1 ·

Basic reporting

The revised version of the manuscript titled “Users’ needs for a digital smoking cessation application and how to address them: A mixed-methods study” is written in a clear, professional English language. In its current form is fine.

Experimental design

This is an original piece of research that falls within the scope of the journal. The research question is well defined, relevant and meaningful. This piece of research can contribute to fill a knowledge gap. Methodology is appropriate and accurate and follows high technical and ethical standards, and the addition of an a priori calculation of statistical power made it more robust.

Validity of the findings

Data are provided and clearly described for further investigations. Conclusions are clear and well stated, linked to the research questions and based on results.

Additional comments

Overall, this manuscript is well written and reports a well conducted piece of research. In terms of novelty of the content, the manuscript presents interesting preliminary evidence about the possible contribution of digital applications in supporting people to quit smoking. In terms of professional English (expression, grammar and spelling), the manuscript is fine. The journal "PeerJ" is an appropriate scientific venue for this manuscript. In terms of study design and methodology, both are appropriate. Lastly, in terms of ethical issues, I cannot see any significant problem. As the most important criticalities have been solved, I recommend its publication.